# Prognostic Impact of Myosteatosis on Mortality in Hospitalized Patients with COVID-19

**DOI:** 10.3390/diagnostics12092255

**Published:** 2022-09-18

**Authors:** Min-Kyu Kang, Yu-Rim Lee, Jeung-Eun Song, Young-Oh Kweon, Won-Young Tak, Se-Young Jang, Jung-Gil Park, Soo-Young Park

**Affiliations:** 1Department of Internal Medicine, College of Medicine, Yeungnam University, Daegu 42415, Korea; 2Department of Internal Medicine, School of Medicine, Kyungpook National University, Kyungpook National University Hospital, Daegu 41994, Korea; 3Department of Internal Medicine, School of Medicine, Daegu Catholic University, Daegu 42472, Korea

**Keywords:** COVID-19, myosteatosis, visceral adiposity, sarcopenia, body composition

## Abstract

Body composition, including sarcopenia, adipose tissue, and myosteatosis, is associated with unfavorable clinical outcomes in patients with coronavirus disease (COVID-19). However, few studies have identified the impact of body composition, including pre-existing risk factors, on COVID-19 mortality. Therefore, this study aimed to evaluate the effect of body composition, including pre-existing risk factors, on mortality in hospitalized patients with COVID-19. This two-center retrospective study included 127 hospitalized patients with COVID-19 who underwent unenhanced chest computed tomography (CT) between February and April 2020. Using the cross-sectional CT images at the L2 vertebra level, we analyzed the body composition, including skeletal muscle mass, visceral to subcutaneous adipose tissue ratio (VSR), and muscle density using the Hounsfield unit (HU). Of 127 patients with COVID-19, 16 (12.6%) died. Compared with survivors, non-survivors had low muscle density (41.9 vs. 32.2 HU, *p* < 0.001) and high proportion of myosteatosis (4.5 vs. 62.5%, *p* < 0.001). Cox regression analyses revealed diabetes (hazard ratio [HR], 3.587), myosteatosis (HR, 3.667), and a high fibrosis-4 index (HR, 1.213) as significant risk factors for mortality in patients with COVID-19. Myosteatosis was associated with mortality in hospitalized patients with COVID-19, independent of pre-existing prognostic factors.

## 1. Introduction

Coronavirus disease (COVID-19), declared a pandemic on 11 March 2020, has increased mortality rates (2.3% to 7.2%) and socioeconomic burden worldwide [1,2]. In South Korea, the COVID-19 outbreak occurred in Daegu and Gyeongsangbuk-do provinces in March 2020. We conducted a multicenter cohort study examining the mortality of 1005 hospitalized patients with COVID-19 from 20 February to 14 April 2020; the fibrosis-4 index (FIB-4), diabetes mellitus (DM), systemic inflammatory response syndrome (SIRS) on admission, and low lymphocyte count were found to be relevant prognostic factors for mortality in patients with COVID-19, which is consistent with the results of several other studies [3,4,5,6,7,8]. In addition, comorbidities, including hypertension, chronic obstructive pulmonary disease, cardiovascular disease, and liver cirrhosis, are known to be major risk factors for COVID-19 [1,9,10].

Obesity, characterized by surplus adipose tissue deposition (visceral adiposity) and muscle fat accumulation (myosteatosis), has been found to be an independent prognostic factor for mortality in patients with COVID-19 [11,12,13]. However, the body mass index (BMI)-based obesity definition is limited because it does not accurately reflect skeletal muscle and fat distribution [14]. Body composition, including skeletal muscle, adipose tissue, and myosteatosis, can be measured using a cross-sectional image at a specific level on computed tomography (CT) [15,16].

Several studies have shown that sarcopenia is a predictor of worse clinical outcomes, including progression of moderate to severe COVID-19, prolonged hospital stay, and COVID-19 syndrome [17,18,19]. In addition, high visceral adiposity and myosteatosis are associated with unfavorable clinical outcomes, including death, transition to severe conditions, and prolonged hospital stays in patients with COVID-19 [12,16,20,21]. Myosteatosis, also known as intramuscular adipose tissue (IMAT), reflects low muscle quality and contributes to progressive muscle damage due to chronic insults of proinflammatory cytokines [12].

Few studies have simultaneously adjusted for mortality-related prognostic factors and body composition in patients with COVID-19. The current study aimed to investigate mortality-related body composition factors, including those proven to significantly involve risk, in patients with COVID-19.

## 2. Materials and Methods

### 2.1. Patients and Data Collection

In our previous study, we retrospectively reviewed 1005 hospitalized patients with COVID-19 confirmed by PCR in five tertiary hospitals in Daegu, South Korea, from February to April 2020. Detailed admission criteria based on the regional policy for patients with COVID-19 were identified in a previous publication [7]. A total of 1005 patients with COVID-19 were reclassified based on the following criteria: (i) patients from three institutions who did not undergo chest CT (*n* = 856) and (ii) patients with inadequate anthropometric variable data (*n* = 22). Finally, 127 patients with COVID-19 who underwent unenhanced chest CT at the time of admission at two tertiary hospitals were enrolled in the current study (Figure 1). The end date for mortality follow-up was August 2020.

A complete description of medical records, including anthropometric, epidemiological, laboratory data, treatments, and clinical outcomes in accordance with WHO interim guidance, has been previously published [7].

The requirement for informed consent was waived, owing to the retrospective nature of the study. The study protocol was approved by the Institutional Review Board of Yeungnam University Hospital (IRB No. 2020-04-060).

### 2.2. Assessment of Body Composition

Using a Picture Archiving and Communications System (Centricity, GE Healthcare, Chicago, IL, USA), chest CT was used to evaluate body composition at the second lumbar vertebra (L2), including skeletal muscle mass and visceral and subcutaneous adipose tissue. Unenhanced chest CT images for body composition were measured by a well-validated software (Automated Muscle and Adipose Tissue Composition Analysis) (AutoMATiCA, https://gitlab.com/Michael_Paris/AutoMATiCA; assessed on on 4 January 2022), using a brush tool [22].

Skeletal muscle area (SMA) was estimated as the sum of intra-abdominal muscles, based on standard Hounsfield unit (HU) thresholds of −29 to 150. Visceral adipose tissue area (VATA) was estimated as the fatty tissue area of the inner skeletal muscles, except that of the kidneys, liver, intestines, and other organs, based on the standard HU thresholds of 150 to −50. The subcutaneous adipose tissue area (SATA) was estimated as the boundary of the skeletal muscle and the line of the abdominal skin, based on standard HU thresholds of −190 to −30. Intermuscular adipose tissue (IMAT) was estimated as fat accumulation in the intra-abdominal skeletal muscle, based on standard HU thresholds of −190 to −30 (Figure 2) [16,22].

The indices of body composition were each classified as area (cm^2^) of SMA, VATA, and SATA divided by height squared (m^2^) and were defined as the skeletal muscle index (SMI), visceral adipose tissue index (VATI), and subcutaneous adipose tissue index (SATI), respectively.

Sarcopenia was defined as SMI < 50 cm^2^/m^2^ in men and <39 cm^2^/m^2^ in women [23]. The visceral to subcutaneous adipose tissue ratio (VSR), an indicator of abdominal adipose tissue distribution, was defined as the presence of visceral adiposity [24] at >1.33 for men and >0.71 for women [16]. Myosteatosis was defined as the mean HU of IMAT, <32.7 HU for men and <28.9 HU for women [16].

### 2.3. Statistical Analysis

All continuous data were presented as medians with interquartile ranges and compared using Student’s t-test or the Mann–Whitney U test. Categorical data were presented as a χ^2^ test or Fisher’s exact test, when appropriate. Using the Cox proportional hazards regression model with the backward selection method, the predictive factors of mortality in patients with COVID-19 were assessed, including the significant risk factors, in a previous publication [7]. The relationship between body composition variables and overall survival was determined using the Kaplan–Meier method. To avoid multicollinearity, age, aspartate aminotransferase, alanine aminotransferase, and platelet count were not included as variables of the FIB-4 index in the stepwise multivariable models. Statistical significance was set at *p* < 0.05. All statistical analyses were performed using the R software (version 3.0.2; R Foundation for Statistical Computing, Vienna, Austria).

## 3. Results

### 3.1. Baseline Characteristics

The baseline characteristics of the enrolled patients are summarized in Table 1 and Table 2. Of 127 patients with COVID-19 who underwent chest CT, 16 (12.6%) died in this study. Compared with survivors, non-survivors were older (60.0 vs. 74.5 years, *p* < 0.001), more likely to be men (47.7 vs. 87.5%, *p* = 0.007), and more had DM (15.3 vs. 50.0%, *p* = 0.003). White cell count (5880 vs. 7030 × 10^3^/µL, *p* = 0.034), C-reactive protein (CRP) (1.2 vs. 11.1 mg/L, *p* < 0.001), aspartate aminotransferase (31.0 vs. 48.5 U/L, *p* = 0.004), gamma-glutamyl transferase (24.0 vs. 154.5 U/L, *p* = 0.010), and creatinine kinase levels (63.0 vs. 122.0 U/L, *p* < 0.001) were significantly higher in non-survivors than in survivors. The percentage of those with a low lymphocyte count was higher in survivors than in non-survivors, based on the COV of a previous study (62.2 vs. 6.2%, *p* < 0.001) [25]. However, the percentages of CRP, aspartate aminotransferase, and alanine transferase were not significantly different between survivors and non-survivors.

Regarding treatment and clinical outcomes, relative to survivors, non-survivors were more frequently treated with oxygen therapy (12.6 vs. 62.5%, *p* < 0.001) and continuous renal replacement therapy (0.9 vs. 12.5%, *p* = 0.048) and had higher proportions of intensive care unit (ICU) admission (9.9 vs. 56.2%, *p* < 0.001), septic shock (8.1 vs. 68.8%, *p* < 0.001), ARDS (7.2 vs. 75.0%, *p* < 0.001), and acute kidney injury (2.7 vs. 31.2%, *p* < 0.001). In our cohort, patients with solid and hematological neoplasia were absent.

In the assessment of body composition between survivors and non-survivors, there were no significant differences in values of SMI (37.8 vs. 43.0 cm^2^/m^2^, *p* = 0.358) and VATI (31.6 vs. 48.2 cm^2^/m^2^, *p* = 0.104), but there were significant differences in SATI (42.9 vs. 36.9 cm^2^/m^2^, *p* = 0.035). Compared with survivors, non-survivors had higher VSR (0.6 vs. 1.3, *p* = 0.002), low muscle HU values (41.9 vs. 32.2, *p* < 0.001), and a high proportion of myosteatosis (4.5 vs. 62.5%, *p* < 0.001) except prevalence of sarcopenia (81.1 vs. 81.2%, *p* = 1.000) and visceral adiposity (28.8 vs. 50.0%, *p* = 0.157). There were no significant differences in the duration of hospital stay between survivors and non-survivors (27.0 vs. 23.5 days, *p* = 0.942). All 16 patients died during hospitalization.

### 3.2. Survival Probability According to the Body Composition in Patients with COVID-19

To evaluate the impact of body composition on mortality, we performed survival analysis to evaluate the mortality of patients with COVID-19, according to the presence of sarcopenia, visceral adiposity, and myosteatosis (Figure 3). No significant differences were observed in survival based on the presence of sarcopenia (no sarcopenia: 23.5 days [16.0–30.0] and sarcopenia: 28.0 days [17.5–38.5], *p* = 0.85). Survival in patients with high visceral adiposity was significantly lower than in those with low visceral adiposity (No visceral adiposity: 28.0 days [21.0–37.0] and visceral adiposity: 22.5 days [10.5–35.0], *p* = 0.038). In addition, survival in patients with high myosteatosis was significantly lower than in those with myosteatosis (no myosteatosis: 28.0 days [17.5–34.5] and myosteatosis: 17.0 days [8.5–49.0], *p* < 0.001).

### 3.3. Risk Factors for Mortality in Patients with COVID-19, including Body Composition

In our previous study, we found that a high FIB-4 index, low lymphocyte count, DM, and SIRS were significant risk factors for mortality in patients with COVID-19 receiving respiratory support. Based on these independent risk factors, we performed univariate and multivariate analyses to predict mortality in patients with COVID-19, including body composition. In multivariate analysis, the presence of DM (hazard ratio [HR], 3.587; 95% confidence interval [CI], 1.218–10.562; *p* = 0.02), myosteatosis (HR, 3.667; 95% CI, 1.195–11.250; *p* = 0.023), and a high FIB-4 index (HR, 1.213; 95% CI, 1.067–1.378; *p* = 0.003) were significant risk factors for predicting mortality in patients with COVID-19. The components of the FIB-4 index, including aspartate aminotransferase, alanine aminotransferase, platelets, and age, were not adjusted in the multivariate analysis (Table 3).

### 3.4. Association between Myosteatosis and Mortality in Patients with COVID-19

To evaluate the impact of myosteatosis on predicting mortality in patients with COVID-19, we performed a multivariate analysis using stepwise adjusted models. The presence of myosteatosis was found to be a significant risk factor for predicting mortality in patients with COVID-19, even after adjusting for age and sex (HR: 4.748; 95% CI: 1.432–15.740; *p* = 0.011). In addition, the impact of myosteatosis was maintained after adjustment for age, sex, diabetes status, and presence of SIRS on admission (model 1: HR, 4.585; 95% CI, 1.505–13.963; *p* = 0.007); for lymphocyte count, FIB-4 index, and CRP, inclusive of model 1 (model 2: HR, 3.667; 95% CI, 1.195–11.250; *p* = 0.023); and for body composition including sarcopenia and presence of visceral adiposity, along with the factors of model 2 (model 3: HR, 3.667; 95% CI, 1.195–11.250; *p* = 0.023) (Table 4).

## 4. Discussion

To the best of our knowledge, this is the first study on the impact of myosteatosis on predicting four-month mortality in hospitalized patients with COVID-19 in Korea. In previous COVID-19 studies, myosteatosis was found to be a significant risk factor for predicting mortality, independent of classic risk factors such as DM, the FIB-4 index, and low lymphocyte count [3,4,5,6,7,8,26]. Our study suggests that the measurement of muscle quality using CT could serve as an imaging biomarker for clinical outcomes in hospitalized patients with COVID-19.

Recent studies have found associations between CT-based body composition and clinical outcomes, including severity and mortality, in patients with COVID-19 [16,21,27,28]. However, the evidence of an association between CT-based myosteatosis and mortality is lacking.

Yang et. al. demonstrated that abundant intramuscular fat deposition (IMAT), which is synonymous with myosteatosis, calculated at the L3 level on abdominal CT, was an independent predictor of clinical illness in patients with COVID-19 [16]. Yi et al. reported that myosteatosis measured at the T12 level on chest CT was closely related to the transition of COVID-19 status from mild to severe [29]. Another study demonstrated that a high IMAT index (>2.35) measured at the T12 level on chest CT was associated with high 21-day mortality in patients with COVID-19 [21]. In a meta-analysis, the presence of low muscle quality, defined as low skeletal muscle density, was associated with COVID-19 mortality, despite high heterogeneity [30].

These results are consistent with our findings. However, unlike in previous studies, myosteatosis was associated with mortality in patients with COVID-19 in our study, even after adjusting for anthropometric variables and laboratory parameters associated with the four-month mortality. Some studies have also revealed an association between high adipose fat distribution and unfavorable clinical outcomes in patients with COVID-19. Yang et al. reported that visceral adiposity, defined as the VAT/SAT ratio and myosteatosis, is an independent risk factor associated with critical illness in patients with COVID-19 [16]. Ogata et al. demonstrated that a high VAT/total adipose tissue ratio is an independent risk factor for increased morbidity and mortality in patients with COVID-19 [28].

Conversely, visceral adiposity was not a significant prognostic factor for mortality in the present study. The putative reasons for this are as follows: First, in our study based on the L2 level of chest CT, VAT may have been underestimated compared with that of L3–5, which contains the most visceral adipose tissue [31]. Second, it is possible that the statistical force of visceral adiposity was weakened because of the mortality-related factors associated with COVID-19 included in our study. In a previous study, VAT was a major risk factor for worse clinical outcomes, including ICU admission, in patients with COVID-19, independent of demographic and clinical parameters. However, the *p*-value of the VAT for ICU admission was checked at 0.05 using multivariate analysis (OR, 2.474; CI, 1.017–6.019; *p* = 0.046) [32]. Large-scale studies are required to identify the impact of visceral adipose tissue, independent of the previously identified prognostic factors of COVID-19.

Moreover, some studies have demonstrated an association between high adipose fat distribution and unfavorable clinical outcomes of COVID-19. Yang et al. reported that visceral adiposity, defined as the VAT/SAT ratio and myosteatosis, was an independent risk factor associated with critical illness in COVID-19 [16]. Ogata et al. demonstrated that high levels of VAT/total adipose tissue were independent risk factors for increased morbidity and mortality in patients with COVID-19 [28]. However, visceral adiposity was not a significant prognostic factor for mortality in the present study.

The association between sarcopenia and clinical outcomes in COVID-19 patients remains controversial. Sarcopenia has been closely associated with prolonged hospital stay, including ICU care and/or invasive mechanical ventilation [19,33] and long-standing COVID-19 syndrome [18]. However, a meta-analysis showed no association between SMI and survival rate in patients with COVID-19 [30]. In our study, sarcopenia was not associated with four-month mortality. Studies with larger cohorts are needed to evaluate the impact of sarcopenia in patients with COVID-19.

In summary, myosteatosis as a muscle quality marker is a major risk factor for predicting mortality in patients with COVID-19. The putative mechanism underlying the association between myosteatosis and COVID-19 severity has not yet been fully elucidated. Myosteatosis is closely related to chronic inflammation, including markers such as IL-6, tumor necrosis factor-alpha (TNF-α), and CRP, leading to lower muscle strength and quality of life [34]. Levels of cytokines such as IL-6 and TNF-α increase rapidly, dysregulating the immune response and leading to direct myotoxicity in COVID-19 [16]. Low muscle quality caused by persistent proinflammatory cytokines may contribute to unfavorable clinical outcomes in patients with COVID-19 [35,36]. Although we could not determine the effect of visceral adiposity on mortality in patients with COVID-19, excessive abdominal fat deposition may contribute to the progression of myosteatosis due to persistent insulin resistance and systemic inflammation [37].

This study had several limitations. First, owing to its cross-sectional design, small sample size, and retrospective nature, the causal relationship between body composition and mortality could not be determined. Second, there is a possibility of selection bias, which requires caution when interpreting the results. Chest CT was performed in only two of the five regional tertiary hospitals, which was not adequate to generalize the severity of COVID-19, excluding mortality, and thus, may have led to a selection bias. Third, the measurements of muscle strength and physical status were not evaluated because of the urgent circumstances of COVID-19. However, we measured all body compositions using chest CT, adjusting for significant variables related to increased mortality, as confirmed by our previous research. Fourth, using abdominal CT, L3 is one of the most reliable levels for measuring body composition [38,39]. We measured body composition at the level of L2 using chest CT because of the lack of patient information regarding abdominal CT. One of the reasons for the non-confirmation of visceral adiposity as a significant mortality-related factor could be attributed to L2, which contains a relatively small amount of adipose tissue compared with L3–5 [31,38]. Finally, we assessed the definition of sarcopenia based on the Western population (<50 cm^2^/m^2^ in men and <39 cm^2^/m^2^ in women) [23], which was distinct from observations in a recent Asian population-based study (<36.5 cm^2^/m^2^ in men and <30.2 cm^2^/m^2^ in women) [40]. Despite our study being based on the Asian population, the presence of sarcopenia was insignificant in both the univariate and multivariate analyses. Further, there are some limitations due to the small number of patients. Nevertheless, the strength of our study is that we identified four-month mortality-related factors in patients with COVID-19, including previously proven prognostic factors as well as body composition.

In conclusion, myosteatosis is associated with mortality in hospitalized patients with COVID-19, independent of other prognostic factors. Therefore, the measurement of body composition may be a potential imaging biomarker for predicting mortality in hospitalized patients with COVID-19.

## Figures and Tables

**Figure 1 diagnostics-12-02255-f001:**
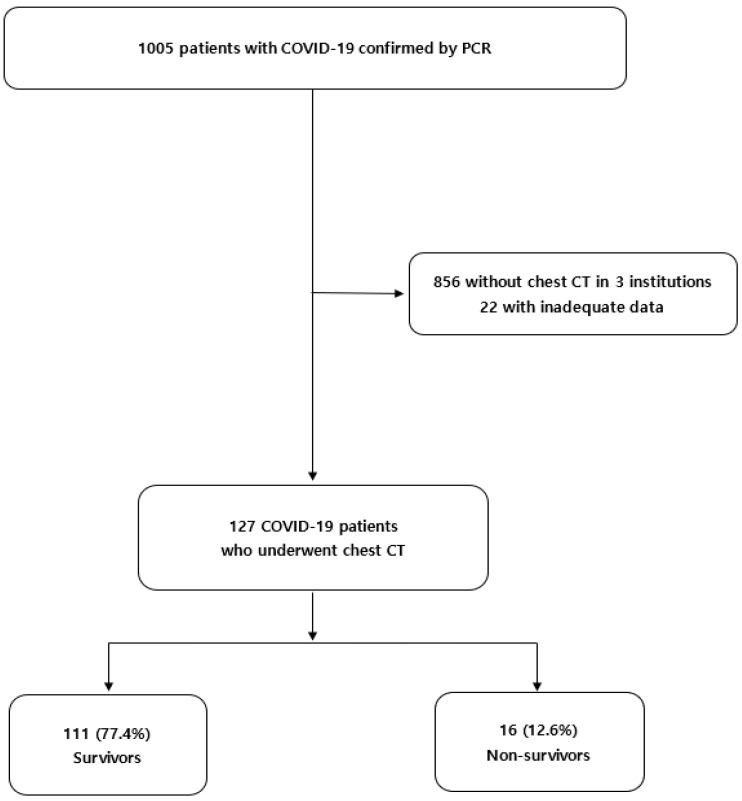
Flow chart of the patients with COVID-19 who underwent chest CT. CT, computed tomography.

**Figure 2 diagnostics-12-02255-f002:**
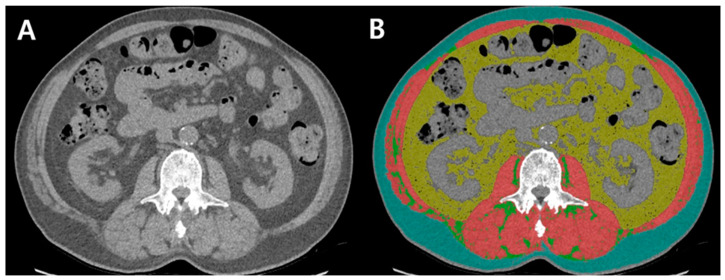
Cross-sectional CT images at the level of L2. (**A**) Unenhanced chest CT, (**B**) using AutoMATiCA software, body composition variables are grouped into the distinct colored area (skeletal muscle area, red; visceral adipose tissue area, yellow; subcutaneous adipose tissue area, blue; intermuscular adipose tissue, green).

**Figure 3 diagnostics-12-02255-f003:**
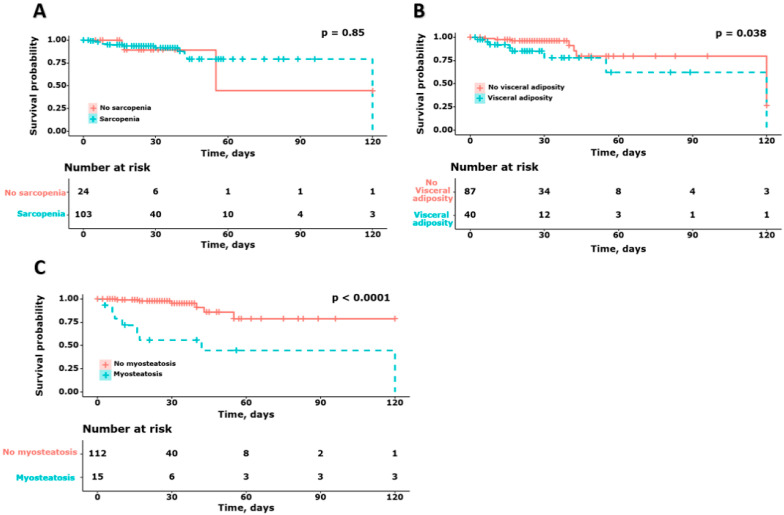
Survival graph according to the presence or absence of sarcopenia (**A**), visceral adiposity (**B**), and myosteatosis (**C**) in patients with COVID-19.

**Table 1 diagnostics-12-02255-t001:** Baseline characteristics of enrolled patients.

Variable	Enrolled Patients *n* = 127
Age (yr)	61.0 [50.0–70.0]
Men, *n* (%)	67 (52.8)
BMI, kg/m^2^	23.6 [21.4–25.4]
**Comorbidities, *n* (%)**	
T2DM	25 (19.7)
Hypertension	46 (36.2)
COPD	7 (5.5)
Chronic kidney disease	4 (3.1)
**Laboratory profiles**	
White cell count, ×10^3^/µL	6000.0 [4690.0–7715.0]
Lymphocyte count, ×10^3^/µL	1209.3 [771.3–1810.1]
Hemoglobin, g/dL	13.1 [12.1–14.0]
Platelet count, ×10^9^/µL	225.0 [158.5–298.0]
C reactive protein, mg/L	2.3 [0.2–10.7]
Aspartate aminotransferase, U/L	32.0 [24.0–47.0]
Alanine aminotransferase, U/L	22.0 [15.0–38.0]
Total bilirubin, mg/dL	0.7 [0.5–1.0]
Gamma glutamyl transferase, U/L	24.0 [17.0–44.0]
Creatinine kinase, U/L	67.0 [46.0–96.5]
Serum ferritin, ng/mL	747.3 [479.9–2023.0]
**Treatments, *n* (%)**	
Oxygen therapy	24 (18.9)
CRRT	3 (2.4)
ECMO	6 (4.7)
**Clinical outcomes, *n* (%)**	
SIRS on admission	32 (25.2)
ICU admission	20 (15.7)
Septic shock	20 (15.7)
ARDS	20 (15.7)
Acute kidney injury	8 (6.3)
**Body composition**	
SMI, cm^2^/m^2^	38.0 [33.1–44.3]
VATI, cm^2^/m^2^	32.0 [20.5–49.8]
SATI, cm^2^/m^2^	41.1 [29.0–56.3]
VSR	0.7 [0.4–1.3]
Muscle HU	41.3 [37.8–44.5]
Sarcopenia, *n* (%)	103 (81.1)
Visceral adiposity, *n* (%)	40 (31.5)
Myosteatosis, *n* (%)	15 (11.8)
Duration of hospital stay, days	27.0 [17.0–36.5]

Values are expressed as median (interquartile range [IQR]) or *n* (%). BMI, body mass index; T2DM, type 2 diabetes mellitus; COPD, chronic obstructive pulmonary disease; CRRT, continuous renal replacement therapy; ECMO, extracorporeal membrane oxygenation; SIRS, systemic inflammatory response syndrome; ICU, intensive care unit; ARDS, acute respiratory distress syndrome; SMI, skeletal muscle index; VATI, visceral adipose tissue index; SATI, subcutaneous adipose tissue index; VSR, visceral-subcutaneous fat ratio; HU, Hounsfield unit.

**Table 2 diagnostics-12-02255-t002:** Baseline characteristics according to survival.

Variable	Survivors *n* = 111 (77.4%)	Non-Survivors *n* = 16 (12.6%)	*p*-Value
Age (yr)	60.0 [46.5–68.5]	74.5 [66.0–79.5]	<0.001
Men, *n* (%)	53 (47.7)	14 (87.5)	0.007
BMI, kg/m^2^	23.6 [21.2–25.4]	22.0 [22.0–27.0]	0.290
**Comorbidities, *n* (%)**			
T2DM	17 (15.3)	8 (50.0)	0.003
Hypertension	39 (35.1)	7 (43.8)	0.695
COPD	4 (3.6)	3 (18.8)	0.218
Chronic kidney disease	2 (1.8)	2 (12.5)	0.127
**Laboratory profiles**			
White cell count, ×10^3^/µL	5880 [4610–7570]	7030 [5940–11,720]	0.034
Lymphocyte count, ×10^3^/µL	1350 [946.4–1836.4]	586.9 [500.0–689.4]	<0.001
Hemoglobin, g/dL	13.1 [12.1–14.2]	12.4 [10.8–13.2]	0.042
Platelet count, ×10^9^/µL	234.0 [164.0–299.5]	165.0 [124.5–268.5]	0.038
C reactive protein, mg/L	1.2 [0.1–7.0]	11.1 [7.6–15.4]	<0.001
Aspartate aminotransferase, U/L	31.0 [23.0–45.0]	48.5 [31.5–63.5]	0.004
Alanine aminotransferase, U/L	21.5 [15.0–37.0]	30.0 [18.0–47.0]	0.412
Total bilirubin, mg/dL	0.7 [0.5–1.0]	0.8 [0.5–0.9]	0.679
Gamma glutamyl transferase, U/L	24.0 [16.0–41.0]	154.5 [75.0–323.0]	0.010
Creatinine kinase, U/L	63.0 [45.0–94.0]	122.0 [94.5–260.0]	0.005
Serum ferritin, ng/mL	620.5 [317.6–1470.6]	868.0 [766.5–2336.5]	0.152
**Treatments, *n* (%)**			
Oxygen therapy	14 (12.6)	10 (62.5)	<0.001
CRRT	1 (0.9)	2 (12.5)	0.048
ECMO	4 (3.6)	2 (12.5)	0.348
**Clinical outcomes, *n* (%)**			
SIRS on admission	25 (22.5)	8 (43.8)	0.128
ICU admission	11 (9.9)	9 (56.2)	<0.001
Septic shock	9 (8.1)	11 (68.8)	<0.001
ARDS	8 (7.2)	12 (75.0)	<0.001
Acute kidney injury	3 (2.7)	5 (31.2)	<0.001
**Body composition**			
SMI, cm^2^/m^2^	37.8 [33.2–43.2]	43.0 [32.7–48.5]	0.358
VATI, cm^2^/m^2^	31.6 [19.9–46.7]	48.2 [21.5–63.0]	0.104
SATI, cm^2^/m^2^	42.9 [29.6–59.5]	36.3 [18.9–45.2]	0.035
VSR	0.6 [0.4–1.3]	1.3 [1.0–1.7]	0.002
Muscle HU	41.9 [38.8–45.2]	32.2 [28.0–37.7]	<0.001
Sarcopenia, *n* (%)	90 (81.1)	13 (81.2)	1.000
Visceral adiposity, *n* (%)	32 (28.8)	8 (50.0)	0.157
Myosteatosis, *n* (%)	5 (4.5)	10 (62.5)	<0.001
Duration of hospital stay, days	27.0 [17.5–33.5]	23.5 [9.0–49.0]	0.942

BMI, body mass index; T2DM, type 2 diabetes mellitus; COPD, chronic obstructive pulmonary disease; CRRT, continuous renal replacement therapy; ECMO, extracorporeal membrane oxygenation; SIRS, systemic inflammatory response syndrome; ICU, intensive care unit; ARDS, acute respiratory distress syndrome; SMI, skeletal muscle index; VATI, visceral adipose tissue index; SATI, subcutaneous adipose tissue index; VSR, visceral-subcutaneous fat ratio; HU, Hounsfield unit.

**Table 3 diagnostics-12-02255-t003:** Cox regression analysis for predicting mortality of COVID-19.

	Univariate Analysis		Multivariate Analysis	
	HR (95% CI)	*p*-Value	HR (95% CI)	*p*-Value
Age	1.066 (1.018–1.115)	0.006		
Male	0.230 (0.051–1.039)	0.056		
BMI	1.204 (1.027–1.411)	0.042		
Type 2 DM	4.054 (1.430–11.494)	0.009	3.587 (1.218–10.562)	0.020
Lymphocyte count, ×10^3^/µL	0.998 (0.997–0.999)	0.003	0.998 (0.997–1.000)	0.065
C reactive protein, mg/L	1.074 (1.1017–1.135)	0.011		
SIRS on admission	3.297 (1.105–9.836)	0.032		
Sarcopenia	0.696 (0.191–2.533)	0.583		
Visceral adiposity	2.070 (0.745–5.753)	0.163		
Myosteatosis	8.182 (2.693–24.859)	<0.001	3.667 (1.195–11.250)	0.023
Fibrosis-4 index	1.286 (1.152–1.436)	<0.001	1.213 (1.067–1.378)	0.003

BMI, body mass index; DM, diabetes mellitus; SIRS, systemic inflammatory response syndrome.

**Table 4 diagnostics-12-02255-t004:** An adjusted hazard ratio of myosteatosis for predicting mortality in patients with COVID-19.

	Myosteatosis	
	HR (95% CI)	*p*-Value
Unadjusted	8.182 (2.693–24.859)	<0.001
Age, sex-adjusted	4.748 (1.432–15.740)	0.011
Multivariate model 1	4.585 (1.505–13.963)	0.007
Multivariate model 2	3.667 (1.195–11.250)	0.023
Multivariate model 3	3.667 (1.195–11.250)	0.023

HR, hazard ratio; CI, confidence interval; SIRS, systemic inflammatory response syndrome; BMI, body mass index. Model 1 was adjusted for age, sex, diabetes, and the presence of SIRS on admission. Model 2 was adjusted for lymphocyte count, Fibrosis-4 index, and C-reactive protein level, inclusive of Model 1. Model 3 was adjusted for BMI, sarcopenia, and visceral adiposity, inclusive of Model 2.

## Data Availability

The data used to support the findings of this study are available from the corresponding author upon request.

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
