# Peer review of "Prognostic Impact of Myosteatosis on Mortality in Hospitalized Patients with COVID-19"

_diagnostics, 2022, doi:10.3390/diagnostics12092255_

Round 1

Reviewer 1 Report

The Authors of the manuscript titled “Prognostic impact of myosteatosis on mortality in hospitalized patients with COVID-19” have reported the results of a retrospective multicentric cohort study that examined the impact of body composition (sarcopenia, visceral and subcutaneous adipose tissue, myosteatosis) evaluated by CT scan on mortality in SARS-CoV-2 infected patients. The Authors observed that myosteatosis was the only body component independently associated with mortality among a cohort of 127 COVID-19 patients hospitalized in South Korea. Several studies showed that increased visceral adiposity and reduced muscle mass are factors associated with poorer outcomes and worse disease severity in COVID-19 patients [Huang Y, et al. Obesity in patients with COVID-19: a systematic review and meta-analysis. Metabolism 2020; 113:154378; Gil S, et al. Muscle strength and muscle mass as predictors of hospital length of stay in patients with moderate to severe COVID-19: a prospective observational study. J Cachexia Sarcopenia Muscle, 2021; 12(6):1871-1878], but few have examined the impact of myosteatosis on mortality.

The study is overall well conducted and the effect of body composition, evaluated at CT scan, on disease progression and severity is an interesting topic to consider. Nevertheless, I summarize several reservations that should be addressed.

Point 1: The Introduction should be developed further. Particularly, the Authors should summarize all the risk factors associated with more severe SARS-CoV-2 infection and not only those found in their previous study. Furthermore, they should report the studies that investigated how sarcopenia impacts COVID-19 and describe better the diagnostic tools for diagnosing myosteatosis, sarcopenia, visceral and subcutaneous adipose tissue.

Point 2: Lines 64-65 the sentence “A total of 1005 patients with COVID-19 were excluded based on the following criteria” is unclear, should be rewritten.

Point 3: Materials and Methods – Based on the differences in body composition between Asians and non-Asians, it seems more practical to use the sarcopenia cut-off for Asians, as recommended by the EWGSOP (Anand A. et al. European Working Group on sarcopenia in older people (EWGSOP2) criteria with population-based skeletal muscle index best predicts mortality in Asians with cirrhosis. J of Clinical and Experimental Hepatology 2022; 12:52-60). Please use a skeletal muscle index < 36.5 cm2/m2 in men and < 30.2 cm2/m2 in women for the analysis and evaluate whether the results obtained remain the same.

Point 4: Materials and Methods – Despite there is no consensus on the myosteatosis cut-off, the Authors chose to use the cut-off used in a previous study carried out on 170 patients. Would it be possible for the Authors to repeat the analysis using the cut-off established in a prior study that involved 569 patients? (Tan L, et al. Diagnosing sarcopenia and myosteatosis based on chest computed tomography images in healthy Chinese adults. Insights Imaging 2021; 12:163).

Point 5: Results – It would be helpful to rework Table 1 and create a table with baseline characteristics of the entire cohort (127 patients) and another table comparing survivors and non-survivors. In terms of comorbidities, the data related to the presence of solid and haematological neoplasia could contribute to defining mortality.

Point 6: Results - Regarding laboratory data, define both linear values and if they are pathological (for example, the numbers of patients with increased transaminases etc).

Point 7: Results – How many deaths occur during hospitalization and after discharge, and what is the percentage? Lines 142-143 put the sentence before the evaluation of body composition.

Point 8: Lines 152-155 “To evaluate the impact of each aspect of body composition on mortality, we performed survival analysis to compare mortality according to the presence or absence of body composition, including sarcopenia, visceral adiposity, and myosteatosis, in patients with COVID-19 receiving respiratory support” rewrite in “To evaluate the impact of body composition on mortality, we performed survival analysis to evaluate mortality of patients with COVID-19 receiving respiratory support according to the presence of sarcopenia, visceral adiposity, and myosteatosis.” Moreover, define the type of respiratory support (orotracheal intubation or any type of oxygen therapy?).

Point 9: Discussion – Provide more details concerning the inflammatory pathway involved in myosteatosis and the possible relationship with COVID-19.

Point 10: English needs more attention. Of the many issues, some examples are listed below: Lines 80-83 “On chest CT using a Picture Archiving and Communications System (Centricity, GE Healthcare), the area of body composition, including skeletal muscle mass and visceral and subcutaneous adipose tissue variables at the level of the second lumbar vertebra (L2), was evaluated” should be rewritten “Using a Picture Archiving and Communications System (Centricity, GE Healthcare), a chest CT was used to evaluate body composition at second lumbar vertebra (L2) including skeletal muscle mass, visceral and subcutaneous adipose tissue” to increase fluidity. The sentence in lines 189-194 is unclear and identifies myosteatosis as a risk factor for the development of diabetes and SIRS etc. Also, the sentence in lines 212-214 is confused. Lines 267-268: IMAT as a synonymous term of myosteatosis was already defined in line 222.

Author Response

The Authors of the manuscript titled “Prognostic impact of myosteatosis on mortality in hospitalized patients with COVID-19” have reported the results of a retrospective multicentric cohort study that examined the impact of body composition (sarcopenia, visceral and subcutaneous adipose tissue, myosteatosis) evaluated by CT scan on mortality in SARS-CoV-2 infected patients. The Authors observed that myosteatosis was the only body component independently associated with mortality among a cohort of 127 COVID-19 patients hospitalized in South Korea. Several studies showed that increased visceral adiposity and reduced muscle mass are factors associated with poorer outcomes and worse disease severity in COVID-19 patients [Huang Y, et al. Obesity in patients with COVID-19: a systematic review and meta-analysis. Metabolism 2020; 113:154378; Gil S, et al. Muscle strength and muscle mass as predictors of hospital length of stay in patients with moderate to severe COVID-19: a prospective observational study. J Cachexia Sarcopenia Muscle, 2021; 12(6):1871-1878], but few have examined the impact of myosteatosis on mortality.

The study is overall well conducted and the effect of body composition, evaluated at CT scan, on disease progression and severity is an interesting topic to consider. Nevertheless, I summarize several reservations that should be addressed.

Point 1: The Introduction should be developed further. Particularly, the Authors should summarize all the risk factors associated with more severe SARS-CoV-2 infection and not only those found in their previous study. Furthermore, they should report the studies that investigated how sarcopenia impacts COVID-19 and describe better the diagnostic tools for diagnosing myosteatosis, sarcopenia, visceral and subcutaneous adipose tissue.

< Answer>

Thank you very much for your meticulous review. Since the contents of comorbidity are not included except for DM, additional contents about comorbidity have been described (page 1, line 42-44). We also described the poor prognosis of Sarcopenia in patients with COVID-19 (page 2, line 52-54). We sincerely appreciate your comments on the rich content structure.

Point 2: Lines 64-65 the sentence “A total of 1005 patients with COVID-19 were excluded based on the following criteria” is unclear, should be rewritten.

<Answer>

It has been replaced with the following phrase: A total of 1005 patients with COVID-19 were reclassified based on the following criteria (page 2, line 70).

Point 3: Materials and Methods – Based on the differences in body composition between Asians and non-Asians, it seems more practical to use the sarcopenia cut-off for Asians, as recommended by the EWGSOP (Anand A. et al. European Working Group on sarcopenia in older people (EWGSOP2) criteria with population-based skeletal muscle index best predicts mortality in Asians with cirrhosis. J of Clinical and Experimental Hepatology 2022; 12:52-60). Please use a skeletal muscle index < 36.5 cm2/m2 in men and < 30.2 cm2/m2 in women for the analysis and evaluate whether the results obtained remain the same.

< Answer >

We sincerely appreciate your valuable comments. Preferentially, we performed statistics using the COVs you mentioned, but it was meaningless in both univariate and multivariate analysis. Presumably, there were some limitations with the small number of patients.

Point 4: Materials and Methods – Despite there is no consensus on the myosteatosis cut-off, the Authors chose to use the cut-off used in a previous study carried out on 170 patients. Would it be possible for the Authors to repeat the analysis using the cut-off established in a prior study that involved 569 patients? (Tan L, et al. Diagnosing sarcopenia and myosteatosis based on chest computed tomography images in healthy Chinese adults. Insights Imaging 2021; 12:163).

< Answer >

Tan et al. defined myosteatosis in the T12 region, and the subjects were healthy controls with an average age of about 43 years of patients. In our study, considering the L2 level and median age (61 years) similar to those of the previous study (L3, 65 years), the COVs were determined despite the small number of patients. Thank you.

Point 5: Results – It would be helpful to rework Table 1 and create a table with baseline characteristics of the entire cohort (127 patients) and another table comparing survivors and non-survivors. In terms of comorbidities, the data related to the presence of solid and haematological neoplasia could contribute to defining mortality.

<Answer>

A table of contents for the entire population has been inserted (page 5-6). Solid and haematological neoplasm were absent in this enrolled patient. Thanks for mentioning something important.

Point 6: Results - Regarding laboratory data, define both linear values and if they are pathological (for example, the numbers of patients with increased transaminases etc).

<Answer>

Thanks for the good comments. We presented meaningful lab comparison values and p values.

White cell count (5880 vs. 7030 ×103/µL, p = 0.034), C-reactive protein (1.2 vs. 11.1 mg/L, p < 0.001), aspartate aminotransferase (31.0 vs. 48.5 U/L, p = 0.004), gamma glutamyl transferase (24.0 vs. 154.5 U/L, p = 0.010), and creatinine kinase levels (63.0 vs. 122.0 U/L, p < 0.001) were significantly higher in non-survivors than in survivors. Regarding treatment and clinical outcomes, relative to survivors, non-survivors were more frequently treated with oxygen therapy (12.6 vs. 62.5 %, p < 0.001)  and continuous renal replacement therapy (0.9 vs. 12.5 %, p = 0.048) and had higher proportions of intensive care unit (ICU) admission (9.9 vs. 56.2 %, p < 0.001), septic shock (8.1 vs. 68.8 %, p < 0.001), ARDS (7.2 vs. 75.0 %, p < 0.001), and acute kidney injury (2.7 vs. 31.2 %, p < 0.001) than survivors (page 4-5, line 134–146).

Point 7: Results – How many deaths occur during hospitalization and after discharge, and what is the percentage? Lines 142-143 put the sentence before the evaluation of body composition.

<Answer>

All 16 patients died during hospitalization. The following content has been inserted in the mentioned area (page 5, line 154).

Point 8: Lines 152-155 “To evaluate the impact of each aspect of body composition on mortality, we performed survival analysis to compare mortality according to the presence or absence of body composition, including sarcopenia, visceral adiposity, and myosteatosis, in patients with COVID-19 receiving respiratory support” rewrite in “To evaluate the impact of body composition on mortality, we performed survival analysis to evaluate mortality of patients with COVID-19 receiving respiratory support according to the presence of sarcopenia, visceral adiposity, and myosteatosis.” Moreover, define the type of respiratory support (orotracheal intubation or any type of oxygen therapy?).

<answer >

We would like to thank the authors for discovering and fixing critical flaws that were previously undiscovered. All patients who did not receive respiratory support were also included in this study. So, we deleted the respiratory support phrase. For reference, respiratory support was any type of oxygen therapy, and out of 24 oxygen therapy patients, 6 mechanical ventilation, 3 HFNC, and 15 nasal/venturi mask apply were performed. Also, it has been modified as mentioned (page 9, line 174-176).

Point 9: Discussion – Provide more details concerning the inflammatory pathway involved in myosteatosis and the possible relationship with COVID-19.

<Answer>

It has been further described as follows. (Page 12, line 290-301)

In summary, myosteatosis as a muscle quality marker is a major risk factor for predicting mortality in patients with COVID-19. The putative mechanism underlying the association between myosteatosis and COVID-19 severity has not been fully elucidated. Myosteatosis, a synonymous term of IMAT, is closely related to chronic inflammation, including IL-6, tumor necrosis factor-alpha (TNF-α), and CRP, leading to lower muscle strength and quality of life in previous study. In particular, cytokines such as IL-6 and TNF-α in-crease rapidly, dysregulating the immune response, leading to direct myotoxicity in COVID-19 status. Low muscle quality caused by persistent proinflammatory cy-tokines may contribute to unfavourable clinical outcomes in patients with COVID-19 [31,32]. Although we could not reveal the effect of visceral adiposity on mortality in patients with COVID-19, excessive abdominal fat deposition may contribute to progression of myosteatosis due to persistent insulin resistance and systemic inflammation.

Point 10: English needs more attention. Of the many issues, some examples are listed below: Lines 80-83 “On chest CT using a Picture Archiving and Communications System (Centricity, GE Healthcare), the area of body composition, including skeletal muscle mass and visceral and subcutaneous adipose tissue variables at the level of the second lumbar vertebra (L2), was evaluated” should be rewritten “Using a Picture Archiving and Communications System (Centricity, GE Healthcare), a chest CT was used to evaluate body composition at second lumbar vertebra (L2) including skeletal muscle mass, visceral and subcutaneous adipose tissue” to increase fluidity. The sentence in lines 189-194 is unclear and identifies myosteatosis as a risk factor for the development of diabetes and SIRS etc. Also, the sentence in lines 212-214 is confused. Lines 267-268: IMAT as a synonymous term of myosteatosis was already defined in line 222.

<Answer>

Thank you very much for your meticulous review. The text you suggested was judged to be more readable and has been modified.

All have been modified as follows.

In addition, it the impact of myosteatosis was maintained after adjustment for diabetes and presence of SIRS on admission inclusive of age and sex (page 10, line 213-215).

In previous studies, myosteatosis was found to be a significant risk factor for predicting mortality (page 11, line 237-238).

Thanks again for your kind and valuable comments. We hope that our revision will meet with approval. We would like to respond to any further questions and comments you may have.

Reviewer 2 Report

The study provided the outcomes of myostasis as a predictor of COVID-19 mortality in Korea in which CT scan data were utilized as a biomarker.

Body composition data from the CT scan represented a proper explanation with reliable segmentation in which each compartment was segmented properly.

Some limitations were mentioned with proper explanations which were decent.

This study provided novel information for myostasis as a predictor for COVID-19 hospitalization using CT scans as imaging biomarkers

Author Response

We appreciate your comprehensive review!

Thank you. 

Round 2

Reviewer 1 Report

Point 1: The Introduction should be developed further. Particularly, the Authors should summarize all the risk factors associated with more severe SARS-CoV-2 infection and not only those found in their previous study. Furthermore, they should report the studies that investigated how sarcopenia impacts COVID-19 and describe better the diagnostic tools for diagnosing myosteatosis, sarcopenia, visceral and subcutaneous adipose tissue.

Answer 1: Thank you very much for your meticulous review. Since the contents of comorbidity are not included except for DM, additional contents about comorbidity have been described (page 1, line 42-44). We also described the poor prognosis of Sarcopenia in patients with COVID-19 (page 2, line 52-54). We sincerely appreciate your comments on the rich content structure.

Comment 1: The Authors provided a revised Introduction.

Point 2: Lines 64-65 the sentence “A total of 1005 patients with COVID-19 were excluded based on the following criteria” is unclear, should be rewritten.

Answer 2: It has been replaced with the following phrase: A total of 1005 patients with COVID-19 were reclassified based on the following criteria (page 2, line 70).

Comment 2: Now the sentence in Lines 70-73 is clearer.

Point 3: Materials and Methods – Based on the differences in body composition between Asians and non-Asians, it seems more practical to use the sarcopenia cut-off for Asians, as recommended by the EWGSOP (Anand A. et al. European Working Group on sarcopenia in older people (EWGSOP2) criteria with population-based skeletal muscle index best predicts mortality in Asians with cirrhosis. J of Clinical and Experimental Hepatology 2022; 12:52-60). Please use a skeletal muscle index < 36.5 cm2/m2 in men and < 30.2 cm2/m2 in women for the analysis and evaluate whether the results obtained remain the same.

Answer 3: We sincerely appreciate your valuable comments. Preferentially, we performed statistics using the COVs you mentioned, but it was meaningless in both univariate and multivariate analysis. Presumably, there were some limitations with the small number of patients.

Comment 3: This point should be stressed as a limitation of the study.

Point 4: Materials and Methods – Despite there is no consensus on the myosteatosis cut-off, the Authors chose to use the cut-off used in a previous study carried out on 170 patients. Would it be possible for the Authors to repeat the analysis using the cut-off established in a prior study that involved 569 patients? (Tan L, et al. Diagnosing sarcopenia and myosteatosis based on chest computed tomography images in healthy Chinese adults. Insights Imaging 2021; 12:163).

Answer 4: Tan et al. defined myosteatosis in the T12 region, and the subjects were healthy controls with an average age of about 43 years of patients. In our study, considering the L2 level and median age (61 years) similar to those of the previous study (L3, 65 years), the COVs were determined despite the small number of patients. Thank you.

Comment 4: For this cohort, Yang et al. cut-off should be more useful.

Point 5: Results – It would be helpful to rework Table 1 and create a table with baseline characteristics of the entire cohort (127 patients) and another table comparing survivors and non-survivors. In terms of comorbidities, the data related to the presence of solid and haematological neoplasia could contribute to defining mortality.

Answer 5: A table of contents for the entire population has been inserted (page 5-6). Solid and haematological neoplasm were absent in this enrolled patient. Thanks for mentioning something important.

Comment 5: Table 1 (baseline characteristics of enrolled patients) was erroneously referred to as Table 2. In my opinion, it is relevant to mention that solid and haematological neoplasia were absent in the entire cohort.

Point 6: Results - Regarding laboratory data, define both linear values and if they are pathological (for example, the numbers of patients with increased transaminases etc).

Answer 6: Thanks for the good comments. We presented meaningful lab comparison values and p values. White cell count (5880 vs. 7030 ×103/µL, p = 0.034), C-reactive protein (1.2 vs. 11.1 mg/L, p < 0.001), aspartate aminotransferase (31.0 vs. 48.5 U/L, p = 0.004), gamma glutamyl transferase (24.0 vs. 154.5 U/L, p = 0.010), and creatinine kinase levels (63.0 vs. 122.0 U/L, p < 0.001) were significantly higher in non-survivors than in survivors. Regarding treatment and clinical outcomes, relative to survivors, non-survivors were more frequently treated with oxygen therapy (12.6 vs. 62.5 %, p < 0.001)  and continuous renal replacement therapy (0.9 vs. 12.5 %, p = 0.048) and had higher proportions of intensive care unit (ICU) admission (9.9 vs. 56.2 %, p < 0.001), septic shock (8.1 vs. 68.8 %, p < 0.001), ARDS (7.2 vs. 75.0 %, p < 0.001), and acute kidney injury (2.7 vs. 31.2 %, p < 0.001) than survivors (page 4-5, line 134–146).

Comment 6: The Authors provided a better explanation for the laboratory data. Might the Authors add data about the normal ranges for each laboratory parameter and the percentage of patients with lab data out of the normal range when comparing survivors and non-survivors?   

Point 7: Results – How many deaths occur during hospitalization and after discharge, and what is the percentage? Lines 142-143 put the sentence before the evaluation of body composition.

Answer 7: All 16 patients died during hospitalization. The following content has been inserted in the mentioned area (page 5, line 154).

Comment 7: All patients died during the hospitalization, according to the Authors.

Point 8: Lines 152-155 “To evaluate the impact of each aspect of body composition on mortality, we performed survival analysis to compare mortality according to the presence or absence of body composition, including sarcopenia, visceral adiposity, and myosteatosis, in patients with COVID-19 receiving respiratory support” rewrite in “To evaluate the impact of body composition on mortality, we performed survival analysis to evaluate mortality of patients with COVID-19 receiving respiratory support according to the presence of sarcopenia, visceral adiposity, and myosteatosis.” Moreover, define the type of respiratory support (orotracheal intubation or any type of oxygen therapy?).

Answer 8: We would like to thank the authors for discovering and fixing critical flaws that were previously undiscovered. All patients who did not receive respiratory support were also included in this study. So, we deleted the respiratory support phrase. For reference, respiratory support was any type of oxygen therapy, and out of 24 oxygen therapy patients, 6 mechanical ventilation, 3 HFNC, and 15 nasal/venturi mask apply were performed. Also, it has been modified as mentioned (page 9, line 174-176).

Comment 8: The Authors clarified the respiratory support used for enrolled patients. As stated in Lines 194-196, a previous study examined the factors associated with mortality in patients with COVID-19 receiving respiratory support. Nevertheless, in this study, patients without respiratory support are enrolled. Did all patients in the previous study receive respiratory support?

Point 9: Discussion – Provide more details concerning the inflammatory pathway involved in myosteatosis and the possible relationship with COVID-19.

Answer 9: It has been further described as follows. (Page 12, line 290-301). In summary, myosteatosis as a muscle quality marker is a major risk factor for predicting mortality in patients with COVID-19. The putative mechanism underlying the association between myosteatosis and COVID-19 severity has not been fully elucidated. Myosteatosis, a synonymous term of IMAT, is closely related to chronic inflammation, including IL-6, tumor necrosis factor-alpha (TNF-α), and CRP, leading to lower muscle strength and quality of life in previous study. In particular, cytokines such as IL-6 and TNF-α in-crease rapidly, dysregulating the immune response, leading to direct myotoxicity in COVID-19 status. Low muscle quality caused by persistent proinflammatory cy-tokines may contribute to unfavourable clinical outcomes in patients with COVID-19 [31,32]. Although we could not reveal the effect of visceral adiposity on mortality in patients with COVID-19, excessive abdominal fat deposition may contribute to progression of myosteatosis due to persistent insulin resistance and systemic inflammation.

Comment 9: The Authors better explain the underlying mechanism involved in myosteatosis and COVID-19.

Point 10: English needs more attention. Of the many issues, some examples are listed below: Lines 80-83 “On chest CT using a Picture Archiving and Communications System (Centricity, GE Healthcare), the area of body composition, including skeletal muscle mass and visceral and subcutaneous adipose tissue variables at the level of the second lumbar vertebra (L2), was evaluated” should be rewritten “Using a Picture Archiving and Communications System (Centricity, GE Healthcare), a chest CT was used to evaluate body composition at second lumbar vertebra (L2) including skeletal muscle mass, visceral and subcutaneous adipose tissue” to increase fluidity. The sentence in lines 189-194 is unclear and identifies myosteatosis as a risk factor for the development of diabetes and SIRS etc. Also, the sentence in lines 212-214 is confused. Lines 267-268: IMAT as a synonymous term of myosteatosis was already defined in line 222.

Answer 10: Thank you very much for your meticulous review. The text you suggested was judged to be more readable and has been modified. All have been modified as follows. In addition, it the impact of myosteatosis was maintained after adjustment for diabetes and presence of SIRS on admission inclusive of age and sex (page 10, line 213-215). In previous studies, myosteatosis was found to be a significant risk factor for predicting mortality (page 11, line 237-238).

Comment 10: The Authors revised the language. 

Author Response

The Authors of the manuscript titled “Prognostic impact of myosteatosis on mortality in hospitalized patients with COVID-19” have reported the results of a retrospective multicentric cohort study that examined the impact of body composition (sarcopenia, visceral and subcutaneous adipose tissue, myosteatosis) evaluated by CT scan on mortality in SARS-CoV-2 infected patients. The Authors observed that myosteatosis was the only body component independently associated with mortality among a cohort of 127 COVID-19 patients hospitalized in South Korea. Several studies showed that increased visceral adiposity and reduced muscle mass are factors associated with poorer outcomes and worse disease severity in COVID-19 patients [Huang Y, et al. Obesity in patients with COVID-19: a systematic review and meta-analysis. Metabolism 2020; 113:154378; Gil S, et al. Muscle strength and muscle mass as predictors of hospital length of stay in patients with moderate to severe COVID-19: a prospective observational study. J Cachexia Sarcopenia Muscle, 2021; 12(6):1871-1878], but few have examined the impact of myosteatosis on mortality.

The study is overall well conducted and the effect of body composition, evaluated at CT scan, on disease progression and severity is an interesting topic to consider. Nevertheless, I summarize several reservations that should be addressed.

Point 1: The Introduction should be developed further. Particularly, the Authors should summarize all the risk factors associated with more severe SARS-CoV-2 infection and not only those found in their previous study. Furthermore, they should report the studies that investigated how sarcopenia impacts COVID-19 and describe better the diagnostic tools for diagnosing myosteatosis, sarcopenia, visceral and subcutaneous adipose tissue.

< Answer>

Thank you very much for your meticulous review. Since the contents of comorbidity are not included except for DM, additional contents about comorbidity have been described (page 1, line 42-44). We also described the poor prognosis of Sarcopenia in patients with COVID-19 (page 2, line 52-54). We sincerely appreciate your comments on the rich content structure.

Comment 1: The Authors provided a revised Introduction.

<Answer > Thank you for your kind review.

Point 2: Lines 64-65 the sentence “A total of 1005 patients with COVID-19 were excluded based on the following criteria” is unclear, should be rewritten.

<Answer>

It has been replaced with the following phrase: A total of 1005 patients with COVID-19 were reclassified based on the following criteria (page 2, line 70).

Comment 2: Now the sentence in Lines 70-73 is clearer.

<Answer > Thank you.

Point 3: Materials and Methods – Based on the differences in body composition between Asians and non-Asians, it seems more practical to use the sarcopenia cut-off for Asians, as recommended by the EWGSOP (Anand A. et al. European Working Group on sarcopenia in older people (EWGSOP2) criteria with population-based skeletal muscle index best predicts mortality in Asians with cirrhosis. J of Clinical and Experimental Hepatology 2022; 12:52-60). Please use a skeletal muscle index < 36.5 cm2/m2 in men and < 30.2 cm2/m2 in women for the analysis and evaluate whether the results obtained remain the same.

< Answer >

We sincerely appreciate your valuable comments. Preferentially, we performed statistics using the COVs you mentioned, but it was meaningless in both univariate and multivariate analysis. Presumably, there were some limitations with the small number of patients.

Comment 3: This point should be stressed as a limitation of the study.

<Answer>

I totally agree with your opinion. The following phrase is inserted in the limitation. Thank you

Finally, we assessed the definition of sarcopenia based on the Western population (< 50 cm2/m2 in men and < 39 cm2/m2 in women) [23], which was distinct from observations in a recent Asian population-based study (< 36.5 cm2/m2 in men and < 30.2 cm2/m2 in women) [40]. Despite our study being based on the Asian population, the presence of sarcopenia was insignificant in both the univariate and multivariate analyses. Further, there are some limitations due to the small number of patients. (page 13, line 309-314)

Point 4: Materials and Methods – Despite there is no consensus on the myosteatosis cut-off, the Authors chose to use the cut-off used in a previous study carried out on 170 patients. Would it be possible for the Authors to repeat the analysis using the cut-off established in a prior study that involved 569 patients? (Tan L, et al. Diagnosing sarcopenia and myosteatosis based on chest computed tomography images in healthy Chinese adults. Insights Imaging 2021; 12:163).

< Answer >

Tan et al. defined myosteatosis in the T12 region, and the subjects were healthy controls with an average age of about 43 years of patients. In our study, considering the L2 level and median age (61 years) similar to those of the previous study (L3, 65 years), the COVs were determined despite the small number of patients. Thank you.

Comment 4: For this cohort, Yang et al. cut-off should be more useful.

<Answer> I agree with your comments. Thank you.

Point 5: Results – It would be helpful to rework Table 1 and create a table with baseline characteristics of the entire cohort (127 patients) and another table comparing survivors and non-survivors. In terms of comorbidities, the data related to the presence of solid and haematological neoplasia could contribute to defining mortality.

<Answer>

A table of contents for the entire population has been inserted (page 5-6). Solid and haematological neoplasm were absent in this enrolled patient. Thanks for mentioning something important.

Comment 5: Table 1 (baseline characteristics of enrolled patients) was erroneously referred to as Table 2. In my opinion, it is relevant to mention that solid and haematological neoplasia were absent in the entire cohort.

<Answer>

I agree your comments, we revised table 1. In addition, we inserted the phrase “In our cohort, the patients with solid and haematological neoplasia were absent” (page 5, line 147) Thank you.

Point 6: Results - Regarding laboratory data, define both linear values and if they are pathological (for example, the numbers of patients with increased transaminases etc).

<Answer>

Thanks for the good comments. We presented meaningful lab comparison values and p values.

White cell count (5880 vs. 7030 ×103/µL, p = 0.034), C-reactive protein (1.2 vs. 11.1 mg/L, p < 0.001), aspartate aminotransferase (31.0 vs. 48.5 U/L, p = 0.004), gamma glutamyl transferase (24.0 vs. 154.5 U/L, p = 0.010), and creatinine kinase levels (63.0 vs. 122.0 U/L, p < 0.001) were significantly higher in non-survivors than in survivors. Regarding treatment and clinical outcomes, relative to survivors, non-survivors were more frequently treated with oxygen therapy (12.6 vs. 62.5 %, p < 0.001)  and continuous renal replacement therapy (0.9 vs. 12.5 %, p = 0.048) and had higher proportions of intensive care unit (ICU) admission (9.9 vs. 56.2 %, p < 0.001), septic shock (8.1 vs. 68.8 %, p < 0.001), ARDS (7.2 vs. 75.0 %, p < 0.001), and acute kidney injury (2.7 vs. 31.2 %, p < 0.001) than survivors (page 4-5, line 134–146).

Comment 6: The Authors provided a better explanation for the laboratory data. Might the Authors add data about the normal ranges for each laboratory parameter and the percentage of patients with lab data out of the normal range when comparing survivors and non-survivors?

<Answer >

Thanks for mentioning.

The percentage of those with a low lymphocyte count was higher in survivors than in non-survivors, based on the COV of a previous study (62.2 vs. 6.2 %, p < 0.001) [25]. However, the percentages of CRP, aspartate aminotransferase, and alanine transferase were not significantly different between survivors and non-survivors. has been attached. (Page 5, line 137-141)

Point 7: Results – How many deaths occur during hospitalization and after discharge, and what is the percentage? Lines 142-143 put the sentence before the evaluation of body composition.

<Answer>

All 16 patients died during hospitalization. The following content has been inserted in the mentioned area (page 5, line 154).

Comment 7: All patients died during the hospitalization, according to the Authors.

<Answer > Yes, thank you.

Point 8: Lines 152-155 “To evaluate the impact of each aspect of body composition on mortality, we performed survival analysis to compare mortality according to the presence or absence of body composition, including sarcopenia, visceral adiposity, and myosteatosis, in patients with COVID-19 receiving respiratory support” rewrite in “To evaluate the impact of body composition on mortality, we performed survival analysis to evaluate mortality of patients with COVID-19 receiving respiratory support according to the presence of sarcopenia, visceral adiposity, and myosteatosis.” Moreover, define the type of respiratory support (orotracheal intubation or any type of oxygen therapy?).

<answer >

We would like to thank the authors for discovering and fixing critical flaws that were previously undiscovered. All patients who did not receive respiratory support were also included in this study. So, we deleted the respiratory support phrase. For reference, respiratory support was any type of oxygen therapy, and out of 24 oxygen therapy patients, 6 mechanical ventilation, 3 HFNC, and 15 nasal/venturi mask apply were performed. Also, it has been modified as mentioned (page 9, line 174-176).

Comment 8: The Authors clarified the respiratory support used for enrolled patients. As stated in Lines 194-196, a previous study examined the factors associated with mortality in patients with COVID-19 receiving respiratory support. Nevertheless, in this study, patients without respiratory support are enrolled. Did all patients in the previous study receive respiratory support?

<Answer >

In the previous study, 289 (28.8%) of 1005 patients received respiratory support, and in this study, the group using chest CT was extracted from 1005 patients separately, so not all patients received respiratory support. DM, Fib-4, SIRS, and low lymphocytes were identified as risk factors for COVID-19 in several papers as well as our existing papers, so we wanted to reveal new risk factors including body composition. Our previous studies are as follows. Clin Mol Hepatol. 2020 Oct;26(4):562-576. & BMJ Open. 2020 Nov 12;10(11):e041989.) Thank you.

Point 9: Discussion – Provide more details concerning the inflammatory pathway involved in myosteatosis and the possible relationship with COVID-19.

<Answer>

It has been further described as follows. (Page 12, line 290-301)

In summary, myosteatosis as a muscle quality marker is a major risk factor for predicting mortality in patients with COVID-19. The putative mechanism underlying the association between myosteatosis and COVID-19 severity has not been fully elucidated. Myosteatosis, a synonymous term of IMAT, is closely related to chronic inflammation, including IL-6, tumor necrosis factor-alpha (TNF-α), and CRP, leading to lower muscle strength and quality of life in previous study. In particular, cytokines such as IL-6 and TNF-α in-crease rapidly, dysregulating the immune response, leading to direct myotoxicity in COVID-19 status. Low muscle quality caused by persistent proinflammatory cy-tokines may contribute to unfavourable clinical outcomes in patients with COVID-19 [31,32]. Although we could not reveal the effect of visceral adiposity on mortality in patients with COVID-19, excessive abdominal fat deposition may contribute to progression of myosteatosis due to persistent insulin resistance and systemic inflammation.

Comment 9: The Authors better explain the underlying mechanism involved in myosteatosis and COVID-19.

<Answer > Thank you for your meticulous review.

Point 10: English needs more attention. Of the many issues, some examples are listed below: Lines 80-83 “On chest CT using a Picture Archiving and Communications System (Centricity, GE Healthcare), the area of body composition, including skeletal muscle mass and visceral and subcutaneous adipose tissue variables at the level of the second lumbar vertebra (L2), was evaluated” should be rewritten “Using a Picture Archiving and Communications System (Centricity, GE Healthcare), a chest CT was used to evaluate body composition at second lumbar vertebra (L2) including skeletal muscle mass, visceral and subcutaneous adipose tissue” to increase fluidity. The sentence in lines 189-194 is unclear and identifies myosteatosis as a risk factor for the development of diabetes and SIRS etc. Also, the sentence in lines 212-214 is confused. Lines 267-268: IMAT as a synonymous term of myosteatosis was already defined in line 222.

<Answer>

Thank you very much for your meticulous review. The text you suggested was judged to be more readable and has been modified.

All have been modified as follows.

In addition, it the impact of myosteatosis was maintained after adjustment for diabetes and presence of SIRS on admission inclusive of age and sex (page 10, line 213-215).

In previous studies, myosteatosis was found to be a significant risk factor for predicting mortality (page 11, line 237-238).

Comment 10: The Authors revised the language.

<Answer>

We revised the language. Thank you.

Thanks again for your kind and valuable comments. We hope that our revision will meet with approval. We would like to respond to any further questions and comments you may have.
